# Effects of gastrocnemius functional massage on lower extemity spasticity, spatio- temporal gait variables and fall risk in patients with stroke: A randomized controlled trial

**Aziz Dengiz[1], Serbay Sekeroz[2]\*, Emre Baskan[3], Güzin Kara[3]**

**1** Muş Alparslan University, Faculty of Health Sciences, Department of Physiotherapy and Rehabilitation, Muş, Türkiye, **2** Van Yüzüncü Yıl University, Faculty of Health Sciences, Department of Physiotherapy and Rehabilitation, Van, Türkiye, **3** Pamukkale University, Faculty of Physiotherapy and Rehabilitation, Denizli, Türkiye

\* serbaysekeroz@gmail.com

## Abstract

### Aim

This randomized controlled clinical trial investigated the effects of gastrocnemius functional massage (GFM) combined with neurodevelopmental treatment (NDT) on spasticity, gait parameters, and functional mobility in stroke patients.

### Methods

A total of 28 chronic stroke survivors were randomized into an experimental group (EG, n = 13) and a control group (CG, n = 15). Both groups received NDT twice a week for six weeks, while the EG received additional GFM. Spasticity (Modified Ashworth Scale), gait parameters (LegSys), and functional mobility (Timed Up and Go test) were assessed pre- and post-treatment.

### Results

The results showed significant improvements in spasticity within the EG for the hip adductor (p = 0.002), knee extensor (p = 0.006), and ankle plantar flexor muscles (p = 0.002), compared to minimal changes in the CG (p > 0.05). Gait analysis revealed significant improvements in the EG for stride number (p = 0.0001), stride length (p = 0.006), stride time (p = 0.001), and stride velocity (p = 0.002), whereas the CG showed no significant changes (p > 0.05). Functional mobility improvements in the EG included reduced sit-to-stand time (p = 0.021) and total Timed Up and Go time (p = 0.001), indicating enhanced dynamic balance and lower extremity strength.

**Data availability statement:** All relevant data are within the manuscript and its Supporting Information files.

**Funding:** The author(s) received no specific funding for this work.

**Competing interests:** The authors have declared that no competing interests exist.

## Conclusion

These findings suggest that combining GFM with NDT significantly enhances spasticity reduction, gait parameters, and functional mobility in stroke patients. Future studies are needed to explore the long-term effects and underlying mechanisms of this combined approach.

## Clinical trial registration

This study was registered at www.clinicaltrials.gov under the identification number NCT06265753.

## Introduction

Spasticity is a common complication following a stroke, affecting approximately 43.2% of patients within the first year [1]. Among chronic stroke survivors with moderate to severe motor impairments, elevated muscle tone (spasticity) is observed in as many as 97% of cases [2]. From a biomechanical perspective, increased gastrocnemius muscle tone can exert excessive plantar force on the foot and ankle joints. Prolonged plantar positioning of the ankle may lead to joint dysfunction and restricted mobility, particularly in the talocrural joint. This condition often contributes to a range of deformities, such as equinovarus and toe deformities [3,4]. Inadequate ankle range of motion, combined with altered gastrocnemius and soleus muscle tone, results in impaired mobility, including asymmetric weight-bearing on the paretic leg, reduced gait speed, decreased cadence, and an increased risk of falls. Consequently, chronic stroke patients often experience significant limitations in daily activities [5–7].

A wide range of rehabilitation interventions has been employed to address post-stroke spasticity, improve gait function, and enhance balance. These include calf muscle stretching, strength training, weight-bearing exercises, orthoses, spasticity management techniques (e.g., botulinum toxin injections, medications), functional electrical stimulation training, proprioceptive control exercises, joint mobilization with movement, passive joint mobilization, and soft tissue massage [5,8,9]. Manual therapy approaches including soft tissue mobilization and joint mobilization manage tone problems such as spasticity through different mechanisms. These mechanisms are; providing non-noxious afferent input to regulate spinal reflex excitability, improving muscle-tendon viscoelastic properties, and providing proprioceptive input to increase quality movement [10–12].

Within this context, functional massage (FM) is a manual therapy technique that combines passive rhythmic mobilization with massage-stretching to reduce excessive muscle tone without causing pain [10]. Studies have shown that various mobilization techniques applied to the ankle joint post-stroke can improve spasticity, balance, gait, and reduce fall risk [11–13]. However, research specifically examining the effects of functional massage on the gastrocnemius muscle remains limited. Instrument-Assisted Soft Tissue Mobilization has shown potential to improve muscle tone and range of motion in conditions such as cerebral palsy and stroke [14,15].

Nevertheless, no studies to date have investigated the effects of functional massage targeting the gastrocnemius muscle on spasticity, gait metrics, and fall risk in stroke patients. The primary aim of this study was to examine the effects of functional massage applied to the gastrocnemius muscle twice a week over six weeks on spasticity, spatiotemporal gait features, and fall risk in stroke patients.

It was hypothesized that functional massage would significantly improve spasticity, gait parameters, and reduce fall risk in stroke patients.

## Methods

### Study design

We carried out this randomized controlled clinical trial in Physical Therapy Department at Pamukkale University Hospital. The local ethics committee of the university—Research Ethics Committee was approved with (E-60116787-020-474196) number—according to the Declaration of Helsinki (World Medical Association, Ferney-Voltaire, France, 2013), and was registered at www.clinicaltrials.gov under the identification number NCT06265753. The CONSORT 2010 standards were adhered to.

### Participants

This study comprised a total of 28 participants who had experienced a stroke, (13 experimental group (EG), 15 control group (CG)). The participants were required to meet specific criteria in order to be included in the study. These criteria included: 1) having experienced a stroke at least 6 months prior, 2) being able to stand independently for at least 1 minute, 3) scoring at least 24 on the Standardised Mini-Mental State Examination (SMMSE), indicating no significant cognitive impairment [16], and 4) having lower extremity spasticity rated between 1–3 on the modified Asworth scale 5) at least 18 years of age. We excluded patients with comorbidities such as neurological, orthopaedic, metabolic, and rheumatological diseases that could potentially impact soft tissues. Additionally, we excluded patients who had received spasticity or other treatments that could affect soft tissues within the past 3 months, those with lower extremity injuries, and those with a diagnosis of cancer or who were pregnant. Participants were informed about the study procedure while a family member was present, and written informed consent was obtained from the participants.

### Procedure

Participant recruitment and treatments of the study was conducted between May 2 and November 29, 2024. The group allocations by simple randomization was created using the website https://www.randomizer.org/ (Fig 1).

Both groups received NDT twice a week for six weeks, with the experimental group receiving 10 minutes of Gastrocnemius function massage (GFM) and the control group receiving 10 minutes of GFM sham application. The NDT was tailored to the patient's needs and clinical status and lasted approximately 45 minutes.

### Gastrocnemius Function Massage (GFM)

Functional Massage (FM) is a massage technique that combines joint movements with targeted pressure on tissues to address muscle and tendon pain as well as joint function limitations [17]. During FM application, pressure is applied to the tendons and soft tissues surrounding the muscles, while the muscles are alternately shortened and lengthened, resulting in joint movement. FM is commonly used to treat soft tissue adhesions, chronic and acute muscle tightness, muscle guarding reflexes, edema, swelling, and pain [17].

GFM was applied to reduce gastrocnemius muscle spasticity and to increase foot and ankle mobility in the participants. During the GFM application, the patient was positioned prone on an appropriate treatment bed. The patient's foot was fixed between the therapist's hip and ilium. The gastrocnemius muscle was grasped with both hands, and the ankle was

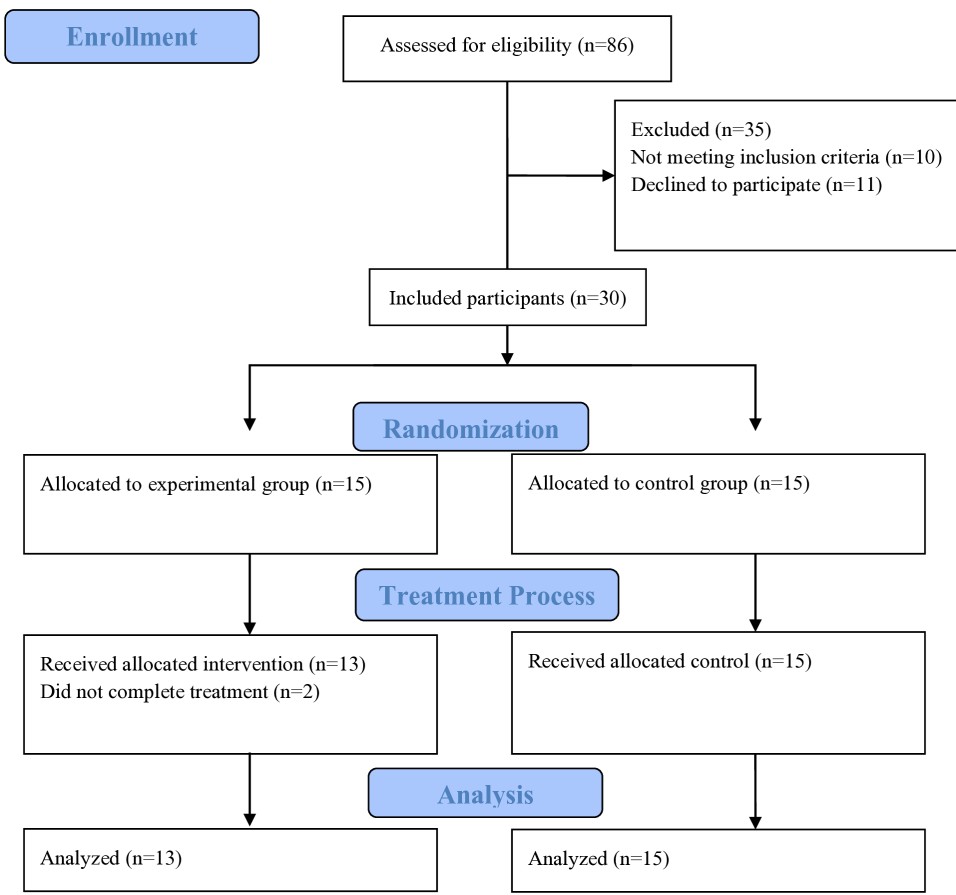

**Fig 1. Flowchart of the Study.**

pushed into dorsiflexion using support from the therapist's trunk. Simultaneously, mobilisation was applied in the cranial direction, parallel to the gastrocnemius muscle fibres, with both hands. Finally, the therapist's arm and trunk were gently pulled backwards until relaxation of the foot and gastrocnemius muscle was achieved. This application is continued rhythmically. The sham application was performed with the patient in the prone position with the hands touching the gastrocnemius muscle without any movement. GFM was performed by a physiotherapist with 11 years of experience in osteopathy and manual therapy.

### Gastrocnemius sham application

The patient lay prone on the therapy table with the foot supported by a towel to maintain a slight flexion position. The therapist gently placed their hands on the gastrocnemius muscle without applying any pressure or stressing the tissues, maintaining this position throughout the procedure.

### Neurodevelopmental Treatment (Bobath therapy) (NDT)

NDT, commonly referred to as the Bobath concept, is the most popular and frequently employed method for rehabilitating hemiparetic patients in Europe. Following injury to the central nervous system, NDT attempts to restore function. The goal of the NDT concept is to acquire complete independence in everyday living activities [18]. In order to optimise functionality,

Bobath provides intensive exercise programmes that are integrated into daily life. These programmes focus on many aspects such as balance, gait, posture, muscle strength, and coordination. NDT emphasises movement quality, precision, task-orientation, intensity, and patient-orientation [19]. In this study, we used the NDT method in accordance with the patients' needs for balance, weight transfer, trunk rotation exercises, upper extremity exercises, stair activities, and fine motor skill exercises as well as walking exercises (weight transfer, stepping, side walking, cross walking, backward walking, etc.). A physiotherapist with 11 years of experience and training in the current Bobath method carried out the NDT.

### Outcome measurements

We elicted and documented participants' demographic data, using a self-designed self-report form, and we assessed participant spasticity using Modified asworth scale (MAS), participant GP using spatial temporal gait analysis (LEGSystm), and participant fall risk using timed up and go test (TUG). We conducted these assessments twice, both pre and post treatments.

### Modified Asworth Scale (MAS)

We used MAS to asses spasticity level of participants. The MAS is a tool used to evaluate the level of resistance encountered when performing passive range of motion, which does not require any instrumentation and is quick to perform (Bohannon and Smith, 1987). It is widely utilised to assess the effectiveness of pharmaceutical and rehabilitative therapies in treating and managing spasticity in patients with Stroke [20].

The scale is represented by the following numerical values [20].

0: No incrase in muscle tone

1: There is a small increase in muscular tension, with a slight resistance or moderate difficulty at the end of the range of motion when moving an affected body part(s) either flexion or extension.

1+: A small increase in muscle tone is followed by minimal resistance across the remaining (less than half) of the range of motion.

2: There is a significant rise in muscular tension across most of the range of motion, but the affected part(s) can still be moved with ease.

3: Significant increase in muscle tone; difficulty with passive movement

4: Affected part(s) rigid in flexion or extension

The MAS (0, 1, 1+, 2, 3, 4) was produced by Bohannon and Smith in 1987. "0, 1, 2, 3, 4, 5" scoring is employed for MAS in a number of investigations [21,22].

We recorded the spasticity levels of the subjects' hemiparetic quadriceps femoris, hip adductor, and gastrocnemius muscles. Furthermore, the overall MAS score was calculated for the affected lower extremity by aggregating all the data. The overall agreement was very good, with a kappa value of 0.87 [23].

### Spatial Temporal Gait Analysis (Legsystm)

We evaluated the participants' walking variables with a wearable walking analysis device called Legsystm. The data collected by two sensors attached to supramalleolar area with velcro is communicated and analysed to the accompanying computer via bluetooth. Legsystm has the capability to conduct various measurements of spatio-temporal data collected from participants. In Legsystm analysis, we asked participants to complete the timed up and go test [24,25].

### Timed up and Go Test (TUG) with legsystm

We employed the Timed Up and Go (TUG) test to assess the spatio-temporal parameters and determine the risk of falling in stroke patients. During the TUG test, the patient is first seated in a suitable chair. They are then directed to stand up,

walk a specified distance of 3 metres, and then return to the chair.. Stroke patients with balance impairments are at a significantly elevated risk of falling, with the prevalence of falls reaching up to 73% [26].

## Sample size

We used G*Power 3.19 (Heinrich Heine University, Dusseldorf, Germany) for sample size calculations [27]. Based on a reference study reporting mean (±SD) changes of 0.877±0.799 in the treatment group and −0.279±0.729 in the control group, the estimated effect size (Cohen's d) was calculated as approximately 1.51 [25]. Using G*Power, a sample size of 11 participants per group was determined to achieve 95% power at a 5% significance level. To account for an anticipated ~25% dropout rate, we planned to recruit 15 participants per group, resulting in a total sample of 28 participants with 2 drop out from control group.

## Statistical analysis

Statistical analyses were performed using the Statistical Package for the Social Sciences (SPSS) version 23.0 (SPSS Inc., Chicago, IL). The normality of all continuous variables was assessed using the Shapiro-Wilk test as well as skewness and kurtosis values. Since the assumptions of parametric tests were not satisfied, the Mann-Whitney U test was applied for comparisons between groups, and the Wilcoxon Signed-Rank test was used for within-group comparisons. The level of significance was set at $p < 0.05$.

## Results

The demographic and clinical data of the groups were presented in Table 1. Participants in both groups were similar in terms of age, height, weight, BMI, stroke duration, type, affected side, and disability level ($p > 0.05$).

Intra-group and inter-group comparisons of spasticity values were presented in Table 2. The current study showed that the EG significantly reduced spasticity in hip adductor muscles ($p = 0.002$), knee extensor muscles ($p = 0.006$), and ankle plantar flexor muscle ($p = 0.002$) while CG reduced in hip adductor muscles ($p = 0.018$). Furthermore, comparison of

**Table 1. Comparison of demographic and clinical values of stroke patients.**

|  | EG (n=13) Median (Min-Max) | CG (n=15) Median (Min-Max) | z | p |
|---|---|---|---|---|
| Age (year) | 54.0 (30–68) | 55.0 (30–73) | −0.116 | 0.908 |
| Height (cm) | 165 (150–175) | 170 (147–175) | −0.791 | 0.429 |
| Weight (kg) | 73 (60–86) | 70 (63–82) | −0.488 | 0.626 |
| BMI (kg/m²) | 26.06 (22.04–31.25) | 26.12 (22.86–31.93) | −0.161 | 0.872 |
| Stroke Duration (month) | 34 (5–72) | 45 (5-120) | −1.041 | 0.298 |
| Modified Rankin Scale | 2 (1–3) | 2 (2–3) | −0.984 | 0.325 |
|  | n (%) | n (%) | df | p |
| Stroke type Ischemic Hemorrhagic Embolic | 11 (84.6) 1 (7.7) 1 (7.7) | 11 (73.3) 3 (20) 1 (6.7) | 2 | 0.650 |
| Affected side Right Left | 7 (53.8) 6 (46.2) | 5 (33.3) 10 (66.7) | 1 | 0.274 |

EG: Experimental group, CG: Control group n: Number of cases, Min-Max: Minimum-Maximum, z: Mann Whitney U test, kg/m²: Kilogram/square meter, BMI: Body mass index, $p < 0.05$.

**Table 2. Comparison of spasticity values of stroke patients.**

| | | EG (n=13) Median (Min-Max) | CG (n=15) Median (Min-Max) | IQR (EG/CG) | z | pᵃ |
|---|---|---|---|---|---|---|
| **Hip adductor muscles** | **Pre-Treatment** | 3 (2-4) | 3 (1-3) | (1.00/1.00) | −.387 | 0.699 |
| | **Post-Treatment** | 2 (1-2) | 2 (1-3) | (1.00/1.00) | −1.174 | 0.240 |
| | **Delta (Δ)** | 1 | 0 | (0.00/1.00) | −1.824 | 0.068 |
| | **w** | 66 | 28 | | | |
| | **pᵇ** | **0.002** | **0.018** | | | |
| **Knee extensor muscles** | **Pre-Treatment** | 1 (1-2) | 2 (0-3) | (1.00/1.00) | −0.104 | 0.917 |
| | **Post-Treatment** | 2 (1-3) | 2 (0-3) | (1.00/1.00) | −1.556 | 0.12 |
| | **Delta (Δ)** | 1 (0-2) | 0 (−1-2) | (1.00/1.00) | −1.725 | 0.085 |
| | **w** | 45 | 22 | | | |
| | **pᵇ** | **0.006** | 0.187 | | | |
| **Ankle Plantar Flexor muscle** | **Pre-Treatment** | 3 (2-4) | 3 (1-4) | (1.00/1.00) | −0.352 | 0.725 |
| | **Post-Treatment** | 1 (1-2) | 2 (1-3) | (1.00/1.00) | −2.741 | **0.006** |
| | **Delta (Δ)** | 1 (0-2) | 0 (−1-2) | (1.00/1.00) | −3.064 | **0.002** |
| | **w** | 78 | 21 | | | |
| | **pᵇ** | **0.002** | 0.24 | | | |

EG: Experimental group, CG: Control group, IQR: Interquartile range, Min-Max: Minimum-Maximum, pᵃ: Intergroup comparison, pᵇ: Intragroup comparison, z: Mann Whitney U test, w: Wilcoxon Signed Rank test, p<0.05.

changes after treatment revealed that spasticity in the ankle plantar flexor muscle decreased significantly more in EG than in CG (p=0.002).

Intra-group comparisons of gait parameters showed that stride number (p=0.002), stride length (p=0.005), stride time (p=0.002), and cadence (p=0.004) were statistically changed in EG, but there was no statistical change in the CG. Additionally, comparison of changes after treatment revealed that stride length (p=0.006), stride velocity (p=0.002), and cadence (p=0.002) increased and stride number (p=0.0001) and stride time (p=0.001) decreased significantly more in EG compared to CG (see Table 3).

Across multiple components of the Timed up and Go test, sit to stand (p=0.021), walk out (p=0.025), mid turn (p=0.001), end turn (p=0.004), and total time (p=0.001) were statistically decreased in EG, but there was no statistical improvement in the CG. Comparison of the changes obtained after treatment in both groups showed that mid turn (p=0.001), walk back (p=0.002) and total time (p=0.001) decreased significantly more in EG than in CG (see Table 4).

## Discussion

The present study demonstrated that gastrocnemius functional massage applied to stroke patients in the experimental group resulted in significant improvements in spasticity, gait parameters, and functional mobility compared to the control group. Moreover, incorporating GFM into NDT further enhanced rehabilitation outcomes, indicating that the combination of GFM and NDT may offer greater effectiveness in post-stroke recovery.

Spasticity, a common consequence of stroke, negatively affects functional characteristics of individuals such as walking, personal care, mobility, and transfer [28]. In our study, the reductions in muscle tone, especially in the hip adductor and ankle plantar flexor muscles, were significant in EG. The post-treatment improvements in MAS scores indicate that adding GFM to the NDT effectively reduced spasticity. Tone problems, which impedes the range of motion and functional activities such as walking and transferring, was notably reduced, aligning with previous studies that highlight the effectiveness of manual therapy interventions in spasticity management [12,29–31]. Parikh et al. [32] showed that manual

**Table 3. Comparison of basic gait parameters of stroke patients.**

| | | EG (n=13) Median (Min-Max) | CG (n=15) Median (Min-Max) | IQR (EG-CG) | z | pᵃ |
|---|---|---|---|---|---|---|
| **Stride number** | **Pre-Treatment** | 12 (6-32) | 11 (5-18) | (7 - 4.5) | −0.786 | 0.432 |
| | **Post-Treatment** | 9 (5-15) | 11 (5-20) | (4 - 4.5) | −1.945 | 0.052 |
| | **Delta (Δ)** | 3 (0-19) | 0 (−7−4) | (3 - 1.5) | −3.646 | **0.0001** |
| | **w** | 78 | 15 | | | |
| | **pᵇ** | **0.002** | 0.218 | | | |
| **Stride length** | **Pre-Treatment** | 0.6 (0.07-1.24) | 0.55 (0.33-1.16) | (0.13 - 0.155) | −1.153 | 0.249 |
| | **Post-Treatment** | 0.76 (0.38-1.44) | 0.57 (0.33-1.2) | (0.15 - 0.21) | −2.005 | **0.045** |
| | **Delta (Δ)** | −0.13 (−0.31-0.11) | 0.00 (−0.7-0.16) | (0.1–0.095) | −2.766 | **0.006** |
| | **w** | 4.5 | 55 | | | |
| | **pᵇ** | **0.005** | 0.90 | | | |
| **Stride time** | **Pre-Treatment** | 1.44 (1.06–2.3) | 1.43 (1.05–1.82) | (0.29 - 0.275) | −0.392 | 0.695 |
| | **Post-Treatment** | 1.2 (0.99–1.71) | 1.5 (0.28–1.8) | (0.28–0.32) | −2.813 | **0.005** |
| | **Delta (Δ)** | 0.25 (−0.05–1.10) | −0.02 (−0.45–1.08) | (0.12–0.13) | −3.25 | **0.001** |
| | **w** | 89.5 | 41 | | | |
| | **pᵇ** | **0.002** | 0.49 | | | |

EG: Experimental group, CG: Control group, IQR: Interquartile range, Min-Max: Minimum-Maximum, pᵃ: Intergroup comparison, pᵇ: Intragroup comparison, z: Mann Whitney U test, w: Wilcoxon Signed Rank test, p<0.05.

| | | EG (n=13) Median (Min-Max) | CG (n=15) Median (Min-Max) | IQR (EG-CG) | z | pᵃ |
|---|---|---|---|---|---|---|
| **Stride velocity** | **Pre-Treatment** | 0.45 (0.04-0.85) | 0.39 (0.18-1.10) | (0.14 −0.17) | −1.014 | 0.311 |
| | **Post-Treatment** | 0.6 (0-0.98) | 0.36 (0.16-1.0) | (0.11–0.20) | −2.282 | **0.023** |
| | **Delta (Δ)** | −0.15 (−0.37-0.45) | 0.00 (−0.1-0.17) | (0.15–0.08) | −3.044 | **0.002** |
| | **w** | 18 | 62.5 | | | |
| | **pᵇ** | 0.059 | 0.248 | | | |
| **Cadence** | **Pre-Treatment** | 83.8 (53–114) | 86 (66 - 114) | (18–16.8) | −0.230 | 0.818 |
| | **Post-Treatment** | 98 (70.3–122) | 88 (70–110) | (10–21) | −2.051 | **0.04** |
| | **Delta (Δ)** | −11 (−27−4) | −0.84 (−12.1–16) | (18.2–5.15) | −3.135 | **0.002** |
| | **w** | 2 | 48 | | | |
| | **pᵇ** | **0.004** | 0.801 | | | |

EG: Experimental group, CG: Control group, IQR: Interquartile range, Min-Max: Minimum-Maximum, pᵃ: Intergroup comparison, pᵇ: Intragroup comparison, z: Mann Whitney U test, w: Wilcoxon Signed Rank test, p<0.05.

therapy—specifically myofascial release—can significantly reduce spasticity and increase joint mobility in stroke patients. Similarly, joint mobilizations targeted at enhancing lower limb mobility resulted in better functional outcomes [12,31]. These results are supported by our findings, especially the decrease in ankle plantar flexor spasticity, which suggests that manual therapy techniques targeting the soft tissues may play a critical role in reducing spasticity. Our results suggest that adding manual interventions (gastrocnemius function massage) to NDT may have specifically targeted the neural mechanisms underpinning muscle tone regulation. Future studies could examine the mechanistic pathways by which manual therapies affect spasticity and guide more effective rehabilitation plans.

Stroke patients often display compensatory gait patterns due to muscle spasticity, impaired balance, and weakness. Notable improvements in gait parameters highlight the potential of our intervention to improve ambulatory function. Post-treatment improvements in stride length and velocity in our study group suggest enhanced motor

**Table 4. Comparison of timed up and go test times of stroke patients.**

| | | EG (n = 13) Median (Min-Max) | CG (n = 15) Median (Min-Max) | IQR (EG – CG) | z | pª |
|---|---|---|---|---|---|---|
| **Sit to stand (sec)** | **Pre-Treatment** | 5.44 (3–7.16) | 5.9 (4–8.49) | (1.37–2.52) | −0.898 | 0.369 |
| | **Post-Treatment** | 4 (3–5.47) | 5.53 (3–8) | (1.49–1.83) | −2.653 | **0.008** |
| | **Delta (Δ)** | 0.94 (−0.81 −4.16) | 0.0 (−3.5–3.35) | (1.6–0.875) | −1.221 | 0.222 |
| | **w** | 78 | 49 | | | |
| | **pᵇ** | **0.021** | 0.456 | | | |
| **Walk out (sec)** | **Pre-Treatment** | 6.00 (3.75–29.8) | 5 (1.04–13.2) | (2.75–3.89) | −1.060 | 0.289 |
| | **Post-Treatment** | 4.73 (3–11.1) | 4.71 (1.60 - 18) | (1.36 - 4.86) | −0.3 | 0.764 |
| | **Delta (Δ)** | 2 (−1.61- 18.7) | 0.09 (−14–2.67) | (3.64 - 1.13) | −1.728 | 0.084 |
| | **w** | 68 | 56 | | | |
| | **pᵇ** | **0.025** | 0.485 | | | |
| **Mid turn (sec)** | **Pre-Treatment** | 4.38 (2.2–14.5) | 3.8 (0–8.76) | (1.36–1) | −1.129 | 0.259 |
| | **Post-Treatment** | 3 (2- 5.36) | 3.8 (1–8.7) | (1.56 − 1) | −1.775 | 0.076 |
| | **Delta (Δ)** | 1.1 (−0.36–9.38) | −0.01 (−1–1.41) | (1.01–0.395) | −3.480 | **.001** |
| | **w** | 89 | 37 | | | |
| | **pᵇ** | **0.001** | 0.576 | | | |
| **End turn (sec)** | **Pre-Treatment** | 1.72 (1.0–6.0) | 2.3 (0.62–5.0) | (1.13–4.93) | −0.968 | 0.333 |
| | **Post-Treatment** | 1.5 (0.86–4) | 2.06 (1.0–5.68) | (1.48–3.62) | −1.246 | 0.213 |
| | **Delta (Δ)** | 0.25 (−2.21–3.97) | 0.12 (−3.0–0.98) | (1.64–1.51) | −0.392 | 0.695 |
| | **w** | 87 | 50 | | | |
| | **pᵇ** | **0.004** | 0.78 | | | |

EG: Experimental group, CG: Control group, IQR: Interquartile range, Min-Max: Minimum-Maximum, pª: Intergroup comparison, pᵇ: Intragroup comparison, z: Mann Whitney U test, w: Wilcoxon Signed Rank test, sec: Second, p<0.05.

| | | EG (n = 13) Median (Min-Max) | CG (n = 15) Median (Min-Max) | IQR (EG – CG) | z | pª |
|---|---|---|---|---|---|---|
| **Walk back (sec)** | **Pre-Treatment** | 5.53 (4.0–32.1) | 5.55 (2.0–11.8) | (1.13–4.93) | −0.553 | 0.58 |
| | **Post-Treatment** | 3.53 (2.67–8.41) | 6.50 (2.0–12.0) | (1.48–3.62) | −1.867 | 0.062 |
| | **Delta (Δ)** | 2.0 (−1.22–23.7) | 0.0 (−9.0–2.24) | (1.64–1.51) | −3.157 | **0.002** |
| | **w** | 63 | 70 | | | |
| | **pᵇ** | 0.244 | 0.286 | | | |
| **Stand to sit (sec)** | **Pre-Treatment** | 3.11 (0.61–4.10) | 2.0 (0.5–6.12) | (2.07–1.67) | −1.359 | 0.174 |
| | **Post-Treatment** | 1.66 (0.15–5.0) | 1.69 (0.5–7.0) | (1.50–1.68) | −0.553 | 0.58 |
| | **Delta (Δ)** | 0.20 (−1.56–3.25) | −0.05 (−2–2.39) | (2.39–0.58) | −1.590 | 0.112 |
| | **w** | 67 | 32.5 | | | |
| | **pᵇ** | 0.146 | 0.638 | | | |
| **Total (sec)** | **Pre-Treatment** | 27.1 (16.4–94.0) | 24.0 (9.0–51.2) | (10.0–10.5) | −0.806 | 0.42 |
| | **Post-Treatment** | 19.8 (13.6–29.0) | 26.0 (14.6–49.4) | (5.6–9.85) | −2.257 | **0.024** |
| | **Delta (Δ)** | 6.44 (−2.74–65.0) | 0.88 (−25.4–6.21) | (6.23–3.56) | −3.325 | **0.001** |
| | **w** | 89 | 61.5 | | | |
| | **pᵇ** | **0.001** | 0.594 | | | |

EG: Experimental group, CG: Control group, IQR: Interquartile range, Min-Max: Minimum-Maximum, pª: Intergroup comparison, pᵇ: Intragroup comparison, z: Mann Whitney U test, w: Wilcoxon Signed Rank test, p<0.05.

control. Manual therapy techniques, including soft tissue mobilization, are known to improve joint mobility and muscle flexibility, which can improve gait patterns. The research by Cho and Park [12] showed that joint mobilization can enhance the joint range of motion and function in stroke patients, resulting in better gait performance. Similarly, previous research [33] has shown that rehabilitation strategies targeting soft tissue can significantly enhance functional gait performance in stroke patients. Furthermore, improved muscle coordination, which has been demonstrated to benefit from manual therapies, may be related to the beneficial effects on cadence seen in the current study [34,35]. The decrease in stride time observed in our results, consistent with the study by Kim et al. [36], may be related to increased dynamic balance ability and reduced compensation during walking. By addressing underlying biomechanical dysfunctions, GFM techniques that enhance joint and muscle function are important to improve gait parameters.

Our results of the TUG test showed significant improvements in functional mobility in the study group, with a significant reduction in total TUG time. This situation demonstrates increased motor efficiency and coordination, which are critical for daily functions such as transfer, turning, and walking. The decrease in sit-to-stand time in our study group suggests increased dynamic balance and lower extremity strength. The beneficial results seen in our study are similar to studies showing that manual therapy techniques such as myofascial release and joint mobilizations significantly increase functional mobility [25,37]. Additionally, the increase in mid-turn time in the current study points to improved dynamic stability and postural control. Manual therapy techniques targeting the soft tissues around the pelvis, spine, and lower extremities have been shown to increase postural control during dynamic functions like turning [38].

The strength of our study is that it was conducted as a randomized and controlled study. Naturally, like any study, ours has some limitations. First, the long-term effectiveness of the treatments was not evaluated. Second, no blinding was implemented in this study.

The conclusion of this study highlights the potential of adding gastrocnemius function massage to NDT as an effective treatment option for improving spasticity, gait parameters, and functional mobility in stroke patients. These improvements may have significant clinical implications, including enhanced independence, reduced risk of fall, and improved quality of life for stroke survivors. Further studies that will optimize the intervention protocol and evaluate its cost-effectiveness will be beneficial for the rehabilitation process of stroke patient.

## Supporting information

**S1 Data. CONSORT Checklist: The checklist provides detailed guidance on the essential methodological and reporting items addressed in the study.**
(DOC)

**S2 Text. Study Protocol in English: This document provides the full study protocol in English, detailing the study design, objectives, inclusion and exclusion criteria, procedures, outcome measures, and analysis plan to ensure transparency and reproducibility.**
(DOCX)

**S3 Text. Study Protocol in Turkish: This document provides the full study protocol in Turkish, detailing the study design, objectives, inclusion and exclusion criteria, procedures, outcome measures, and analysis plan to ensure transparency and reproducibility.**
(DOCX)

**S4 Data. Minimal data set: This file contains the minimal anonymized data set underlying the findings of the study, provided in a format that allows independent verification of the analyses presented in the manuscript.** The dataset includes all relevant variables necessary to replicate the results while ensuring participant confidentiality.
(XLSX)

## Acknowledgments

The authors thanks to all participants.

## Author contributions

**Conceptualization:** Aziz Dengiz, Serbay Sekeroz, Güzin Kara.

**Data curation:** Aziz Dengiz, Serbay Sekeroz, Güzin Kara.

**Methodology:** Aziz Dengiz, Emre Baskan, Güzin Kara.

**Supervision:** Emre Baskan.

**Writing – original draft:** Aziz Dengiz, Serbay Sekeroz.

**Writing – review & editing:** Emre Baskan.

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
