## [Decision Letter · Decision Letter 0]

19 Jun 2025

Dear Dr. SEKEROZ,

Thank you for submitting your manuscript to PLOS ONE. After careful consideration, we feel that it has merit but does not fully meet PLOS ONE’s publication criteria as it currently stands. Therefore, we invite you to submit a revised version of the manuscript that addresses the points raised during the review process.

We look forward to receiving your revised manuscript.

Kind regards,

Esedullah Akaras

Academic Editor

PLOS ONE

Journal Requirements:

Reviewers' comments:

Reviewer's Responses to Questions

**Comments to the Author**

1. Is the manuscript technically sound, and do the data support the conclusions?

Reviewer #1: Partly

Reviewer #2: Yes

Reviewer #3: Yes

Reviewer #4: Yes

2. Has the statistical analysis been performed appropriately and rigorously?

Reviewer #1: No

Reviewer #2: Yes

Reviewer #3: Yes

Reviewer #4: Yes

3. Have the authors made all data underlying the findings in their manuscript fully available?

Reviewer #1: Yes

Reviewer #2: Yes

Reviewer #3: Yes

Reviewer #4: Yes

4. Is the manuscript presented in an intelligible fashion and written in standard English?

Reviewer #1: Yes

Reviewer #2: Yes

Reviewer #3: Yes

Reviewer #4: Yes

Reviewer #1: The investigators used G*Power 3.19 (Heinrich Heine University, Dusseldorf, Germany) for sample size calculations. Based on an expected effect size of 1 (statistical power of.80 and error probability of.05), They computed the total estimated required sample size for acceptable statistical power to be 28 (14 in the EG, 14 in the CG). This is insufficient information. What were the precise differences sought with standard deviations, effect sizes, etc? The protocol in the Appendix is insufficient and lacks this information as well. Also, these calculations should be checked against a more standard clinical trials sample size software such as Power and Precision or other relevant tool such as PASS.

The analysis performed looks somewhat reasonable. However, Tables 2 to 4 show many statistical comparisons being made. A p=value adjustment for multiple comparisons should be examined as those like p=0.04 or 0.025 may not be statistically significant. What are the primary and secondary endpoints? How does this affect the setting of the alpha levels for appropriate adjustment? Also, the Statistical Analysis section refers to normal tests. However, the Tables presented look to be done non parametrically. If normality was assumed, then the confidence intervals would be helpful in determining the reliability of the results. Please explain. The paper needs a thorough edit to be sure appropriate detail is provided.

Reviewer #2: 1. It would be useful to include some important demographic details in this section, such as gender distribution, stroke etiology (ischemic/hemorrhagic), and affected side (right/left).

2. GFM Definition: The detailed description of the Gastrocnemius Function Massage (GFM) application is commendable. However, some expressions (for example: “the therapist gently pulls the arm and trunk backwards until the foot and gastrocnemius muscle relaxes”) need clarification in terms of anatomy and biomechanics. Do the terms “arm” and “trunk” here refer to the therapist’s or the patient’s trunk? Unclear.

3. Sham GFM: The “sham” method applied to the control group provides a valid comparison, but the extent to which the procedure was effective in terms of blinding (e.g., whether participants guessed their group) is not specified.

4. Gap: The fact that NDT was administered to all participants suggests that GFM was the only variable; however, more information on session length and degree of individualization in content for NDT would be helpful.

5. MAS: The explanations are sufficient, but it is not stated how many different assessors made the assessments and whether the assessors were blinded.

6. Gait Analysis - Legsystm: Adequate technical explanation is given. However, it would have been appropriate to refer to the validity and reliability information of the device.

7. Deficiency: Taking the effect size as 1 is a very high assumption. It should be stated which source or preliminary study this value is based on.

8. Sentence structures should be simplified; especially the part explaining the GFM application should be written in clearer and more technical terms.

9. A reference to the effect size used in the sample size calculation should be provided.

10. It should be stated whether the person(s) performing the spasticity assessment are the same person and whether blindness is ensured.

11. There are grammar and fluency problems in some sentences. Example: In the sentence “Participants in both groups had similar in terms of…” the expression “had similar” is used incompletely. The correct usage should be “were similar in terms of…”

Reviewer #3: 1. Title & Abstract

Strengths:

The title is clear, specific, and reflects the key outcomes of the study.

The abstract is structured and summarizes aim, methods, results, and conclusion.

Suggestions:

Consider adding "stroke patients" to the keywords for better indexing.

The final sentence about long-term outcomes could be omitted in the abstract (due to word limits) and moved to the conclusion section of the main text.

Introduction

Strengths:

Provides a solid overview of the prevalence and impact of spasticity post-stroke.

Clear rationale and identification of the research gap.

Suggestions:

The hypothesis (H1) should be rephrased in academic style, e.g., “It was hypothesized that functional massage would significantly improve...”.

Methods

Strengths:

Ethical approval and clinical trial registration are well-documented (a strong point).

The sham-controlled design is commendable.

Clear inclusion/exclusion criteria.

Critiques:

Massage description is written in an active, first-person tone (“The therapist stands…”). It should be revised into passive academic tone, e.g., “The therapist was positioned behind the patient…”.

"10 GFM sham application" likely contains a typo — should be “10 minutes of sham GFM”.

MMSE cut-off is listed as 23 — typically, 24 is used. Clarify with reference.

Results

Strengths:

Statistical tests are appropriate.

Tables are referenced, and p-values are reported clearly.

Suggestions:

Verify consistency between text and tables (e.g., changes in cadence or stride parameters).

Discussion

Strengths:

Thorough and well-supported with citations.

Proper alignment of results with existing literature.

Strengths and limitations are acknowledged.

Suggestions:

Some explanations (especially about the mechanism of GFM) are slightly repetitive — could be condensed.

Blinding details are not discussed. Was this single-blind or double-blind?

Language & Style

Critiques:

Several minor spelling errors (e.g., “muslce” → “muscle”, “spacticity” → “spasticity”).

Avoid direct instructions or first-person tone in the methods section.

Overall Assessment

Strengths:

This is a well-designed randomized controlled trial with a novel approach.

Use of a sham group strengthens the internal validity.

Results have meaningful clinical implications.

Weaknesses:

The blinding protocol is unclear.

Final Recommendation

This study provides important insights into post-stroke rehabilitation using manual therapy techniques. With revisions in tone, style, minor methodological clarifications.

Minor revision

Reviewer #4: The manuscript aims to highlight the beneficial effects of functional massage applied in association with neurodevelopmental treatment on lower limb spasticity, within a rehabilitation program focused on improving gait parameters and functional mobility. The study is designed as a randomized and controlled clinical trial, conducted on a group of patients diagnosed with stroke. The results obtained demonstrate the effectiveness of the proposed intervention, supported by the use of standardized, validated and relevant assessment tools for the research objectives. The conclusions of the study highlight the potential of fusing functional massage of the gastrocnemius muscle with neurodevelopmental treatment, as an effective option for reducing spasticity, optimizing gait and increasing functional mobility in stroke patients.

The originality of the study lies in the "merge" of functional gastrocnemius massage with neurodevelopmental treatment, a previously little-explored combination that brings a new and promising perspective in post-stroke recovery, focused on reducing spasticity and optimizing locomotor function.

The title effectively captures the core elements of the study; however, while informative, it is somewhat lengthy and includes a minor inconsistency. Throughout the article, the term "functional massage" is consistently used, whereas the title refers to "function massage." It is unclear whether this variation is intentional or a typographical oversight, but it introduces a degree of ambiguity that could be confusing for the reader.

While the introduction reflects a comprehensive understanding of the clinical dimensions of the pathology and the corresponding therapeutic approaches, it transitions too quickly from a general presentation of treatment methods to highlighting functional massage, without providing a clear transition or a convincing argument regarding the effectiveness of this technique in a neurological context. Although the authors propose functional massage as an effective intervention for reducing spasticity and indicate a promising direction for its merge with neurodevelopmental therapy (NDT), the theoretical foundation remains insufficiently supported. The paper would benefit from a strengthening of the scientific framework by including relevant studies from the specialized literature and by a clearer delimitation of the investigated techniques, which would increase the validity and impact of the conclusions formulated.

Also, the authors don't clearly identify how the study relates to previous published research.

The Methods section offers a thorough overview of participant demographics, inclusion and exclusion criteria, data collection instruments, and statistical procedures. The statistical techniques implemented are appropriate for the nature of the data; even so, there remains room for improvement within this section. In the study period is stated as May–November 2024, and the intervention consisted of twice-weekly therapy sessions for six weeks. It would be useful to clarify the exact time frame: whether each participant received treatment for 6 weeks within this extended period (May–November), or whether the stated period reflects the total duration of the intervention for all participants. This information would help to better understand the organization and delivery of the intervention.

The discussion section accurately reflects the results obtained, with the findings providing strong support for the study’s primary objectives. The manuscript offers a valuable contribution to the stroke rehabilitation literature by shedding light on a relatively underexplored therapeutic approach. Even so, subtle insights can be better appreciated, such as the approach to spasticity which is treated in a relatively general way, with limited emphasis on its manifestations in the lower limb, particularly the foot. This limited focus somewhat diminishes the clarity and specificity of the analysis in relation to the study’s stated objectives. It may be beneficial for the authors to more clearly elaborate on the segmental aspects of spasticity, in order to better support the intervention’s effectiveness on the targeted region.

In addition, the paragraph beginning with "Spasticity, a common consequence of stroke, negatively affects functional characteristics of individuals such as walking, personal care, mobility, and transfer" until the section "Parikh et al. [32] showed that manual therapy—specifically myofascial release—can significantly reduce spasticity and increase joint mobility in stroke patients." contains noticeable redundancy, particularly in the repeated use of the term "spasticity." This excessive repetition makes the text harder to read giving the impression of a hurried or underdeveloped exposition of ideas. The authors are encouraged to revise this section to make the ideas clearer and improve the overall flow of the text.

The conclusion is generally well-aligned with the study’s findings, effectively underscoring the value of promoting a thoughtfully designed and innovative therapeutic approach. It highlights the strengths of the research while also maintaining a balanced perspective by acknowledging its limitations, thereby contributing to the credibility and relevance of the work.

Please check out the "doi" from Reference nr. 15- "Mostafa DE, Olama KA, Aly MG. Effect of instrument-assisted soft tissue mobilization on hamstring flexibility in children with diplegic cerebral palsy. Egypt J Hosp Med.2022;89(1):4842–7. doi:10.21608/ejhm.2022.242276." It seems to direct the reader to a source other than the one indicated.

**Do you want your identity to be public for this peer review?** For information about this choice, including consent withdrawal, please see our Privacy Policy

Reviewer #1: No

Reviewer #2: No

Reviewer #3: No

Reviewer #4: **Yes: ** Lect. Phd. PT. Eva ILIE

---

## [Author Response · Author response to Decision Letter 1]

11 Jul 2025

Reviewer #1:

Dear Reviewer,

Thank you very much for your valuable and constructive comments. They have helped us improve the transparency and scientific rigor of our manuscript. Below we provide point-by-point responses describing the changes made to the text.

1. Sample size and power analysis details

Reviewer comment:

“Based on an expected effect size of 1 ... insufficient information. What were the precise differences sought with standard deviations, effect sizes, etc.? ... These calculations should be checked against a more standard clinical trials sample size software ...”

Response:

Thank you for this important observation. We have now added detailed information about the sample size calculation to the Methods section. In the reference study, the mean change in the treatment group was 0.877 ± 0.799 (mean±SD), while in the control group it was -0.279 ± 0.729 (mean±SD). Based on these data, the calculated Cohen’s d effect size was approximately 1.51. Using G*Power with 95% power (power = 0.95) and a 5% significance level (α = 0.05), the estimated required sample size was 11 participants per group. To account for an anticipated dropout rate of approximately 25%, we planned to recruit additional participants. The study was ultimately completed with a total of 28 participants.

This clarification has been added to the Sample Size Calculation subsection of the Methods.

2. Parametric vs. non-parametric tests and normality assumption

Reviewer comment:

“The Statistical Analysis section refers to normal tests. However, the Tables presented look to be done non parametrically. If normality was assumed ...”

Response:

Thank you for pointing this out. We performed normality testing (Shapiro–Wilk) on our data and found that the assumptions for parametric tests were not satisfied. Therefore, non-parametric tests were used throughout the analysis. This choice is now clearly described in the Statistical Analysis section.

3. Multiple comparisons adjustment

Reviewer comment:

“Tables 2 to 4 show many statistical comparisons ... p-value adjustment for multiple comparisons should be examined ...”

Response:

We appreciate this comment. In our study, the main comparisons involved two independent groups. Since there were no multiple groups or multiple factors tested simultaneously, corrections such as Bonferroni adjustment were not deemed necessary.

Reviewer #2:

Dear Reviewer,

Thank you for your detailed review and constructive feedback on our study. Below, we provide our point-by-point responses to each of your comments:

1. Reviewer comment:

It would be useful to include some important demographic details in this section, such as gender distribution, stroke etiology (ischemic/hemorrhagic), and affected side (right/left).

Response:

The requested information has been provided in Table 1.

2. Reviewer comment:

However, some expressions (for example: “the therapist gently pulls the arm and trunk backwards…”) need clarification in terms of anatomy and biomechanics.

Response:

The noted ambiguity has been resolved, and the text has been revised for clarity.

3. Reviewer comment:

The extent to which the procedure was effective in terms of blinding (e.g., whether participants guessed their group) is not specified.

Response:

It has been clarified that the technique applied to patients was presented as a therapeutic approach, and no participant realized they were receiving a sham procedure.

4. Reviewer comment:

More information on session length and degree of individualization in content for NDT would be helpful.

Response:

The NDT procedure has been described in detail.

5. Reviewer comment:

It is not stated how many different assessors made the assessments and whether the assessors were blinded.

Response:

No blinding was performed. The study was designed and conducted as a randomized controlled trial, and all assessments were carried out by a single researcher.

6. Reviewer comment:

It would have been appropriate to refer to the validity and reliability information of the device.

Response:

Several reference articles regarding the use of the device in stroke patients have been provided.

7. Reviewer comment:

Taking the effect size as 1 is a very high assumption. It should be stated which source or preliminary study this value is based on.

Response:

In the reference study, the mean (±SD) changes were reported as 0.877 ± 0.799 in the treatment group and -0.279 ± 0.729 in the control group. Based on these data, the calculated Cohen’s d effect size was approximately 1.51. Using G*Power with 95% power (power = 0.95) and a 5% significance level (α = 0.05), the estimated required sample size was 11 participants per group. To account for an anticipated ~25% dropout rate, we planned to recruit 14 participants per group. The study was ultimately completed with a total of 28 participants.

8. Reviewer comment:

Sentence structures should be simplified; especially the part explaining the GFM application should be written in clearer and more technical terms.

Response:

Revisions have been made to address these ambiguities and improve clarity.

9. Reviewer comment:

A reference to the effect size used in the sample size calculation should be provided.

Response:

The reference used for the sample size calculation has been added to the manuscript.

10. Reviewer comment:

It should be stated whether the person(s) performing the spasticity assessment are the same person and whether blindness is ensured.

Response:

No blinding was performed. The study was designed and conducted as a randomized controlled trial, and all assessments were carried out by a single researcher.

11. Reviewer comment:

There are grammar and fluency problems in some sentences. Example: In the sentence “Participants in both groups had similar in terms of…” the expression “had similar” is used incompletely. The correct usage should be “were similar in terms of…”.

Response:

Thank you for pointing this out. The relevant sentences have been revised accordingly.

Reviewer #3:

Dear Reviewer,

Thank you very much for your thorough and constructive comments on our manuscript. We have carefully considered each point and revised the text accordingly. Below, we provide point-by-point responses to all of your suggestions:

1. Reviewer comment:

Consider adding "stroke patients" to the keywords for better indexing. The final sentence about long-term outcomes could be omitted in the abstract (due to word limits) and moved to the conclusion section of the main text.

Response:

The phrase “stroke patients” has been added to the keywords. However, to emphasize the importance of this aspect, the phrase “long-term outcomes” was intentionally retained in the abstract.

2. Reviewer comment:

The hypothesis (H1) should be rephrased in academic style, e.g., “It was hypothesized that functional massage would significantly improve...”.

Response:

The hypothesis statement has been revised as suggested to follow academic style.

3. Reviewer comment:

Massage description is written in an active, first-person tone (“The therapist stands…”). It should be revised into passive academic tone, e.g., “The therapist was positioned behind the patient…”. "10 GFM sham application" likely contains a typo — should be “10 minutes of sham GFM”. MMSE cut-off is listed as 23 — typically, 24 is used. Clarify with reference.

Response:

The relevant section has been rewritten in passive voice for academic tone. Information on the duration of GFM (10 minutes) has been added. Regarding MMSE, it was intended to indicate scores greater than 23, but was mistakenly written as “at least 23.” The text has been corrected to specify “at least 24.”

4. Reviewer comment:

Verify consistency between text and tables (e.g., changes in cadence or stride parameters).

Response:

Values for cadence and stride parameters have been checked for consistency between the text and the tables.

5. Reviewer comment:

Blinding details are not discussed. Was this single-blind or double-blind?

Response:

The relevant section has been reviewed, and the absence of blinding has been explicitly added as a limitation.

6. Reviewer comment:

Several minor spelling errors (e.g., “muslce” → “muscle”, “spacticity” → “spasticity”). Avoid direct instructions or first-person tone in the methods section.

Response:

All identified spelling errors have been corrected. The relevant sections have been revised to use passive academic language throughout.

Reviewer #4:

Dear Reviewer,

Thank you very much for your thorough and constructive review of our manuscript. We appreciate the time and effort you dedicated to providing detailed feedback. Below, we address each of your comments point by point.

1. Reviewer comment:

The manuscript aims to highlight the beneficial effects of functional massage applied in association with neurodevelopmental treatment...

Response:

Thank you for your overall positive evaluation and for recognizing the novelty and clinical value of our approach.

2. Reviewer comment:

The title effectively captures the core elements of the study; however, while informative, it is somewhat lengthy and includes a minor inconsistency. Throughout the article, the term "functional massage" is consistently used, whereas the title refers to "function massage."

Response:

Thank you for your careful attention. To eliminate this inconsistency and avoid confusion, we have revised the title to consistently use the phrase “functional massage.”

3. Reviewer comment:

While the introduction reflects a comprehensive understanding of the clinical dimensions... The theoretical foundation remains insufficiently supported...

Response:

We appreciate this valuable observation. We revised the Introduction section to more clearly and thoroughly explain the neurophysiological effects of functional massage. This enhancement improves the clarity and readability of the section and provides a stronger theoretical foundation for the study.

4. Reviewer comment:

It would be useful to clarify the exact time frame: whether each participant received treatment for 6 weeks within this extended period (May–November), or whether the stated period reflects the total duration of the intervention for all participants.

Response:

We have clarified that each participant received treatment for 6 weeks within the overall study period.

5. Reviewer comment:

The paragraph contains noticeable redundancy, particularly in the repeated use of the term "spasticity."

Response:

We have revised the paragraph to reduce repetition, shortened the content to improve clarity, and limited the use of the term “spasticity” for better readability.

6. Reviewer comment:

Please check out the "doi" from Reference nr. 15...

Response:

The incorrect DOI has been corrected to ensure accurate citation.

---

## [Decision Letter · Decision Letter 1]

12 Aug 2025

Dear Dr. SEKEROZ,

Thank you for submitting your manuscript to PLOS ONE. After careful consideration, we feel that it has merit but does not fully meet PLOS ONE’s publication criteria as it currently stands. Therefore, we invite you to submit a revised version of the manuscript that addresses the points raised during the review process.

**ACADEMIC EDITOR: ** After minor corrections, the article becomes suitable for publication.

We look forward to receiving your revised manuscript.

Kind regards,

Esedullah Akaras

Academic Editor

PLOS ONE

Journal Requirements:

Reviewers' comments:

Reviewer's Responses to Questions

**Comments to the Author**

Reviewer #1: (No Response)

Reviewer #2: All comments have been addressed

Reviewer #3: All comments have been addressed

Reviewer #4: All comments have been addressed

2. Is the manuscript technically sound, and do the data support the conclusions?

Reviewer #1: (No Response)

Reviewer #2: Yes

Reviewer #3: Yes

Reviewer #4: Yes

3. Has the statistical analysis been performed appropriately and rigorously?

Reviewer #1: (No Response)

Reviewer #2: Yes

Reviewer #3: Yes

Reviewer #4: Yes

4. Have the authors made all data underlying the findings in their manuscript fully available?

Reviewer #1: (No Response)

Reviewer #2: Yes

Reviewer #3: Yes

Reviewer #4: Yes

5. Is the manuscript presented in an intelligible fashion and written in standard English?

Reviewer #1: (No Response)

Reviewer #2: Yes

Reviewer #3: Yes

Reviewer #4: Yes

Reviewer #1: One can see from the results section that the analyses showed significant changes in the EG group and mostly no significant changes in the CG subjects. The authors should thus tell the reader why no direct p-value was calculated or displayed in the Tables 2 to 4 comparing the pre to post changes in CG vs EG for each result. This could be a limitation of the study and require a larger future trial requiring a more adequate sample size. The Discussion section should be expanded accordingly.

Reviewer #2: I would like to inform you that I found the article you submitted to be scientifically sufficient. Your topic selection, methodological approach, and the contributions of your findings to the field were noteworthy. I believe your work will make valuable contributions to the literature. I thank you for your efforts and wish you continued success.

Reviewer #3: The revisions have been reviewed and the article has been deemed appropriate. I think it is suitable for publication. you.

Reviewer #4: I appreciate the authors' effort to respond rigorously and seriously to the reviewers' comments. All the issues raised have been adequately clarified, including strengthening the theoretical basis in the introduction, standardizing terminology, and detailing the duration of the intervention. The revised version of the manuscript brings significant improvements in terms of clarity, coherence, and scientific quality of the paper. I congratulate the authors for the careful way in which they integrated the feedback and for the obvious concern to perfect the study.

**Do you want your identity to be public for this peer review?** For information about this choice, including consent withdrawal, please see our Privacy Policy

Reviewer #1: No

Reviewer #2: No

Reviewer #3: No

Reviewer #4: **Yes: ** Ilie Eva

---

## [Author Response · Author response to Decision Letter 2]

19 Aug 2025

Reviewer #1: One can see from the results section that the analyses showed significant changes in the EG group and mostly no significant changes in the CG subjects. The authors should thus tell the reader why no direct p-value was calculated or displayed in the Tables 2 to 4 comparing the pre to post changes in CG vs EG for each result. This could be a limitation of the study and require a larger future trial requiring a more adequate sample size. The Discussion section should be expanded accordingly.

Response: We thank the reviewer for this valuable comment. In fact, p-values for both intra-group and inter-group comparisons were calculated and were already presented in Tables 2–4. To make this distinction clearer for readers, we have now added superscript letters a and b next to the p-values, where:

pᵃ: Intergroup comparison

pᵇ: Intragroup comparison

This explanation has been added below the tables to ensure that the distinction between intra-group and inter-group comparisons is clearly understood.

Reviewer #2: I would like to inform you that I found the article you submitted to be scientifically sufficient. Your topic selection, methodological approach, and the contributions of your findings to the field were noteworthy. I believe your work will make valuable contributions to the literature. I thank you for your efforts and wish you continued success.

Response: We are grateful for your positive feedback and encouraging words. Your appreciation of our methodological approach and the potential contribution of our findings is highly motivating for our research team.

Reviewer #3: The revisions have been reviewed and the article has been deemed appropriate. I think it is suitable for publication. you.

Response: We thank you for your positive assessment and for recognizing the adequacy of the revised manuscript. Your supportive comments are greatly appreciated.

Reviewer #4: I appreciate the authors' effort to respond rigorously and seriously to the reviewers' comments. All the issues raised have been adequately clarified, including strengthening the theoretical basis in the introduction, standardizing terminology, and detailing the duration of the intervention. The revised version of the manuscript brings significant improvements in terms of clarity, coherence, and scientific quality of the paper. I congratulate the authors for the careful way in which they integrated the feedback and for the obvious concern to perfect the study.

Response: We greatly appreciate your recognition of the improvements made in the revised manuscript. Your constructive feedback in the earlier review round was invaluable in enhancing the clarity, coherence, and scientific quality of our work.

---

## [Decision Letter · Decision Letter 2]

29 Aug 2025

Effects of Gastrocnemius Functional Massage on Lower Extemity Spasticity, Spatio- Temporal Gait Variables and Fall Risk in Patients with Stroke: A Randomized Controlled Trial

PONE-D-25-03271R2

Dear Dr. SEKEROZ,

We’re pleased to inform you that your manuscript has been judged scientifically suitable for publication and will be formally accepted for publication once it meets all outstanding technical requirements.

Kind regards,

Esedullah Akaras

Academic Editor

PLOS ONE

Additional Editor Comments (optional):

Reviewers' comments:

Reviewer's Responses to Questions

**Comments to the Author**

Reviewer #1: All comments have been addressed

2. Is the manuscript technically sound, and do the data support the conclusions?

Reviewer #1: (No Response)

3. Has the statistical analysis been performed appropriately and rigorously?

Reviewer #1: (No Response)

4. Have the authors made all data underlying the findings in their manuscript fully available?

Reviewer #1: (No Response)

5. Is the manuscript presented in an intelligible fashion and written in standard English?

Reviewer #1: (No Response)

Reviewer #1: (No Response)

**Do you want your identity to be public for this peer review?** For information about this choice, including consent withdrawal, please see our Privacy Policy

Reviewer #1: No

---

## [Editor Report · Acceptance letter]

PONE-D-25-03271R2

PLOS ONE

Dear Dr. Sekeroz,

I'm pleased to inform you that your manuscript has been deemed suitable for publication in PLOS ONE. Congratulations! Your manuscript is now being handed over to our production team.

Kind regards,

on behalf of

Dr. Esedullah Akaras

Academic Editor

PLOS ONE